# Photocatalysis over N-Doped TiO₂ Driven by Visible Light for Pb(II) Removal from Aqueous Media

Endang Tri Wahyuni *[ID], Titi Rahmaniati, Aulia Rizky Hafidzah, Suherman Suherman [ID] and Adhitasari Suratman

Chemistry Department, Faculty of Mathematic and Natural Sciences, Gadjah Mada University, Sekip Utara P.O. Box Bls 21, Yogyakarta 55281, Indonesia; titi.rahmaniati@mail.ugm.ac.id (T.R.); a.rizky.hafidzah@mail.ugm.ac.id (A.R.H.); suherman.mipa@ugm.ac.id (S.S.); adhitasari@ugm.ac.id (A.S.)
* Correspondence: endang_triw@ugm.ac.id; Tel.: +62-274-545188

**Abstract:** The photocatalysis process over N-doped TiO₂ under visible light is examined for Pb(II) removal. The doping TiO₂ with N element was conducted by simple hydrothermal technique and using urea as the N source. The doped photocatalysts were characterized by DRUVS, XRD, FTIR and SEM-EDX instruments. Photocatalysis of Pb(II) through a batch experiment was performed for evaluation of the doped TiO₂ activity under visible light, with applying various fractions of N-doped, photocatalyst mass, irradiation time, and solution pH. The research results attributed that N doping has been successfully performed, which shifted TiO₂ absorption into visible region, allowing it to be active under visible irradiation. The photocatalytic removal of Pb(II) proceeded through photo-oxidation to form PbO₂. Doping N into TiO₂ noticeably enhanced the photo-catalytic oxidation of Pb(II) under visible light irradiation. The highest photocatalytic oxidation of 15 mg/L Pb(II) in 25 mL of the solution could be reached by employing TiO₂ doped with 10%w of N content 15 mg, 30 min of time and at pH 8. The doped-photocatalyst that was three times repeatedly used demonstrated significant activity. The most effective process of Pb(II) photo-oxidation under beneficial condition, producing less toxic and handleable PbO₂ and good repeatable photocatalyst, suggest a feasible method for Pb(II) remediation on an industrial scale.

**Keywords:** doping N; TiO₂; Pb(II); photocatalytic-oxidation; visible light





## 1. Introduction

Lead (Pb), along with Cd, Hg, Cr(VI) and As, is categorized as the most toxic heavy metal [1]. Lead in the form of Pb(II) ion can be present intensively in wastewater of many industries, such as storage batteries, mining, metal plating, painting, smelting, ammunition, oil refining and the ceramic glass [2–7]. Pb(II) ion is non-biodegradable and tends to accumulate in living tissues, causing various diseases and disorders, such as anemia encephalopathy (brain disfunction), cognitive impairment, kidney and liver damage, and toxicity to the reproductive system [2–6]. The permissible limit of Pb(II) in drinking water is 0.005 mg/L according to the current US Environmental Protection Agency (USEPA) standard, while WHO determines the limit is 0.01 mg/L [3]. In fact, the actual concentration of lead in wastewater is as high as several hundred milligram per liter. Therefore, the removal of lead from wastewater before the heavy metal ions contacted with unpolluted natural water bodies is important and urgent.

In recent years, adsorption has become one of the techniques that is frequently applied for Pb(II) removal, because it is a very effective technique in terms of initial cost, simplicity of design, ease of operation and insensitivity to toxic substances [2–6]. Several adsorbents that have been devoted for removal of Pb(II) include palm tree leaves [2], mesoporous activated carbon [3], modified natural zeolite [4], magnetic natural zeolite [5] and sulfonated polystyrene [6]. However, when the adsorbent is saturated with concentrated Pb(II) ions, it becomes solid waste with higher toxicity, which creates new environmental problems.

Currently, remediation of Pb(II) containing water by the photo-oxidation process through photo-Fenton method [7], under $TiO_2$ photocatalyst [8], and photo-assisted electrochemical [9] have also been developed. Furthermore, oxidation of Pb(II) by manganese salt has been studied [10]. Oxidation seems to be the most interesting method since Pb(II) oxidation resulted in the non-toxic precipitate $PbO_2$ and is more easily handled [7–10].

With respect to the photocatalysis process, $TiO_2$ as a photocatalyst has recently received considerable attention for the removal of persistent organic pollutants (POPs), including phenols [11], tetracycline [12], dyes [13] and linier alkyl benzene sulphonate [14] due to its cost-effective technology, non-toxicity, quick oxidation rate and chemical stability [11–26]. The practical application of $TiO_2$ is, however, significantly restricted by its low visible absorption due to its wide bandgap energy (Eg) $\approx$ 3.20 eV [15–26] and immediate charge (electron and hole) recombination [19–22]. With such wide a band gap, $TiO_2$ can only be excited by photons with wavelengths shorter than 385 nm emerging in UV region [15–26]. In fact, UV light only occupies a small portion (about 5%) of the sunlight spectrum [15,19–21,23,24], which limits $TiO_2$ application under low-cost sunlight or visible irradiation. Moreover, the fast charge recombination leads to a less effective photocatalysis process [19–22].

Therefore, an effort has made to overcome these deficiencies by doping $TiO_2$ structure with either metal elements [23–26] or non-metal elements [15–22]. Non-metal dopant has been reported to give be effective in narrowing the gap or decreasing the band gap energy (Eg) of $TiO_2$, compared to the metal dopant. This is because the size of non-metal elements are smaller than the metal dopants [15–22], allowing them to be inserted into the lattice of $TiO_2$ crystal facilely. Among the non-metal dopants that have been examined, N is the most interesting due to the very effective shift of the band gap energy into the lower value [16–21]. It has also been reported that N-doped $TiO_2$ materials exhibit strong absorption of visible light irradiation and significant enhancement of photocatalytic activity [16–24].

The doping N into $TiO_2$ has been performed by sol-gel [16,18,20,21], hydrothermal [21], coprecipitation [19], and plasma-assisted electrolysis [17] methods. Furthermore. according to the related research reports that have been reviewed by Gomez et al. [21], hydrothermal is believed to be the best method, since a large amount of N can be doped, which cannot occur by using other methods. In addition, in this method, the longer time for photocatalyst ageing is not required and the direct using $TiO_2$ powder is possible, suggesting a simple and low-cost process [23]. The hydrothermal method has been employed for doping transition metals [23], but to the best of our knowledge, it has not been explored for N doping.

Many studies of using N-doped $TiO_2$ under visible light for degradation of dyes [17] and para-nitrophenol [18], as well oxidation of $NO_2$ gas [19] and $H_2$ gas production [16], have been used. However a research of using N-doped $TiO_2$ for catalysis of the Pb(II) photo-oxidation and removal under visible irradiation is not traceable.

Under the circumstances, preparation of N-doped$TiO_2$ by hydrothermal method is addressed and photocatalytic activity of N-doped $TiO_2$ is systematically evaluated for photo-oxidation of Pb(II) in the aqueous media driven by visible light. Moreover, the efficiency of the photocatalysis process strongly depends on the operating parameters, including the content of the dopant [16,17,19,20,24], reaction time [12,14,15,23,24], the photocatalyst mass [12,14,24] and the solution pH [8,12,14,24]. Therefore, it is necessary to find the optimal process parameters through laboratory treatability tests. This study is hoped to contribute to the development of doped $TiO_2$ photocatalysts and toxic metal remediation technology.

## 2. Results and Discussion

### 2.1. Characterization of TiO$_2$–N Photocatalyst

#### 2.1.1. By Diffuse Reflectance UV/Vis. (DRUV/Vis.) Method

The DRUV/visible spectra of the photocatalyst samples were displayed as Figure 1. From the spectra, the wavelengths of the absorption edge could be determined that were displayed in Table 1. As expected, doping TiO$_2$ with N atom can shift the absorption into longer wavelength emerging visible region, due to the narrowing their gaps. The narrowing resulted from the insertion of N atom into the lattice of TiO$_2$ crystal. The evident narrowing gap was represented by decreasing band gap energy (Eg) values, as exhibited in Table 1. It is also notable that an increasing amount of introduced N caused Eg to decline more effectively. Some studies have also reported similar findings [15,19,20,24,27].

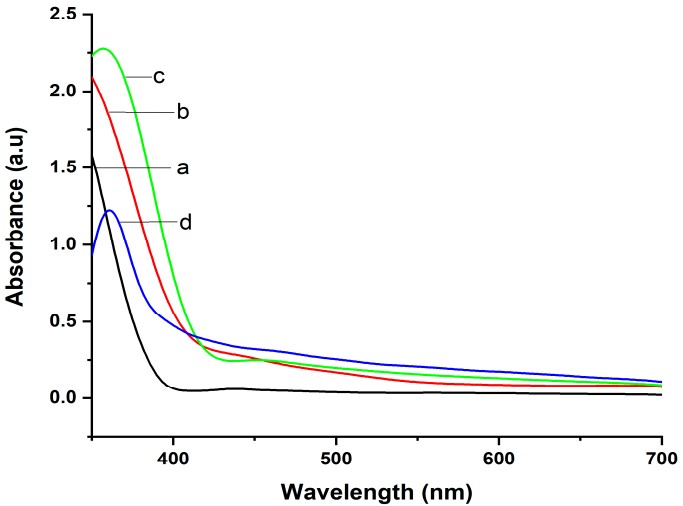

**Figure 1.** The DRUV Spectra of (**a**) TiO$_2$, (**b**) TiO$_2$–N (5) (**c**) TiO$_2$–N (10) and (**d**) TiO$_2$–N (15).

**Table 1.** The effect of N doped into TiO$_2$ on band gap energy.

| Photocatalyst | Wavelength (nm) | Band Gap Energy (eV) |
|:---:|:---:|:---:|
| TiO$_2$ | 387.5 | 3.20 |
| TiO$_2$–N(5) | 405.2 | 3.06 |
| TiO$_2$–N(10) | 411.9 | 3.01 |
| TiO$_2$–N(15) | 418.9 | 2.96 |

The diminution of Eg implied that N atoms have successfully been doped in the TiO$_2$ crystal [19–21]. A doping generally can take place through substitutional and/or interstitial mechanisms and can be distinguished based on the lowering Eg values [27]. According to Ansari et al. [27], the decreasing Eg into 3.06 from 3.20 eV is due to the presence of substitutional mechanism, that is, replacing oxygen of TiO$_2$ by nitrogen dopant; meanwhile, lowering Eg toward less than 2.50 eV assigns to the interstitial mode, referring to the addition of nitrogen atoms into the TiO$_2$ crystal. However, some others [19–21] have been proposed that reduce Eg into 3.0–2.9 eV and can be a form of interstitial doping. Reducing Eg up to 3.06–2.96 eV as observed in this present study reveals that N doping involves a combination of substitutional and interstitial mechanisms [27].

#### 2.1.2. By X-ray Diffraction (XRD) Method

The XRD patterns of TiO$_2$ and TiO$_2$–N photocatalysts are displayed in Figure 2. Several 2θ values of 25.25°, 37.52°, 48.02°, 53.58°, 54.88°, 62.61°, 68.78°, 70.33° and 75.07° are observed, which match with Miller index as (101), (004), (200), (105), (211), (204), (116), (204) and (215) of the lattice planes for anatase TiO$_2$ as listed in JCPDS card no. 21-1272) [16,17,20].

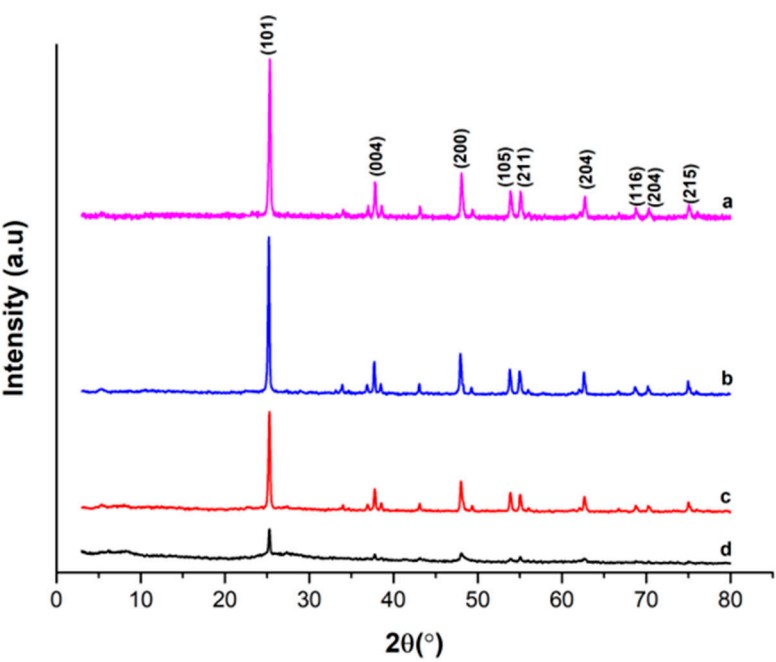

**Figure 2.** The XRD patterns of (**a**) $TiO_2$, (**b**) $TiO_2$–N (5), (**c**) $TiO_2$–N (10) and (**d**) $TiO_2$–N (15).

In the XRD patterns of all N-doped photocatalysts, additional phase, except anatase, is not observed, indicating that the N dopant did not agglomerate to reach the macro level [20]. This finding is in accordance with the result reported previously [16–20].

It is also observable that doping N caused a decrease in the XRD intensities, and the intensities gradually declined with the enhancement of N amount. The decrease of the intensities was partially due to $TiO_2$ crystallinity damage, due to the incorporation of N dopant in the $TiO_2$ lattice [16,21]. The increase of the partially damaged generated more amorphous phase, as demonstrated clearly by the XRD pattern of $TiO_2$–N (15) having highest N content. A similar finding was also delivered by Li et al. [16] and Mahy et al. [18] and was reviewed by Gomez et al. [21]. The slight crystallinity damage to $TiO_2$–N (5) was possible as a result of an interstitial doping [27]. Meanwhile, the more significant damaged as found in $TiO_2$–N (10) and $TiO_2$–N (15) may be caused by substitution of oxygen in the $TiO_2$ lattice by nitrogen dopant atom, or by a simultaneous interstitial and substitutional mechanisms [27]. This XRD data should agree with the trend of Eg decreasing.

### 2.1.3. By Fourier Transform Infra Red (FTIR) Method

In Figure 3, it is apparent that some characteristic bands of urea at the wavenumbers of 3448 $cm^{-1}$ are associated with O-H stretching, 3346 $cm^{-1}$ is related to N-H deformation, 1661 $cm^{-1}$ is assigned to C=O bond, 1604 $cm^{-1}$ originated from N-H ($NH_2$) and 1465 $cm^{-1}$ is due to C-N vibration [28]. Furthermore, the spectra of all $TiO_2$–N samples are similar to those of un-doped $TiO_2$, where several characteristic peaks of $TiO_2$ are observed at around 3400, 1630 and 700–500 $cm^{-1}$ of the wavenumbers. Many studies also reported the same IR spectra [13,15,16,18,21]. The peaks appearing at 3400 were attributed to the Ti–OH bond, a band at ~1630 $cm^{-1}$ to the OH bending vibration of chemisorbed and/or physisorbed water molecule on the surface of $TiO_2$, and the strong band in the range of 700–500 $cm^{-1}$ to stretching vibrations of Ti–O–Ti bond [13,15,16,18,21].

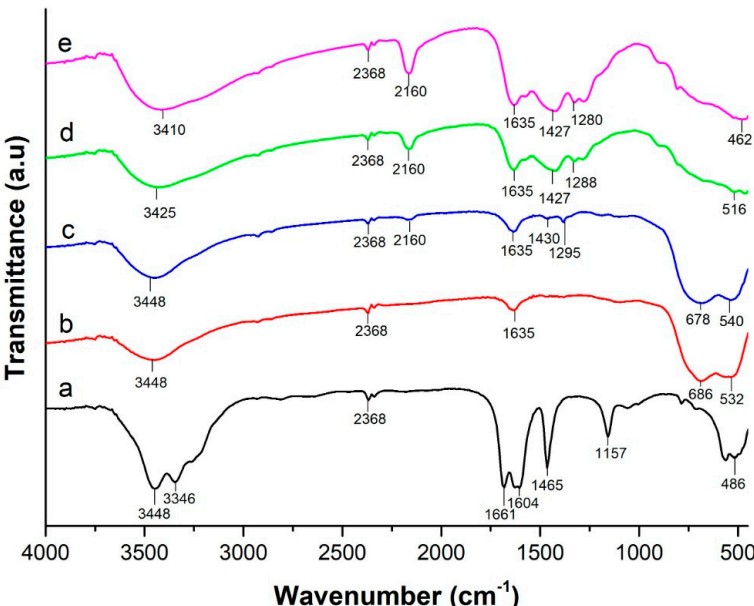

**Figure 3.** FTIR spectra of (**a**) urea, (**b**) $TiO_2$, (**c**) $TiO_2$–N (5), (**d**) $TiO_2$–N (10) and (**e**) $TiO_2$–N (15).

In comparison with the undoped $TiO_2$, the spectra of all $TiO_2$–N samples display additional peaks at around 2160, 1427–1430 and 1288–1295 cm$^{-1}$. The peaks look shaper for $TiO_2$–N with higher N content. The peaks at 1426–1430 and 1270–1290 cm$^{-1}$ were attributed to the vibrations of the N-Ti bond [16,21]. The appearance of the Ni-Ti bond in the samples suggests that N atoms have been incorporated into the $TiO_2$ lattice. The peak at 2155–2160 cm$^{-1}$ can be assigned for carbide ($C_2^{2-}$ or –C≡C–) species that possibly corresponded to the residual of the incompletely decomposed urea [16], during the hydrothermal at 150 °C and calcination at 400 °C. According to Li et al. [16], the complete removal of the carbide species and urea decomposition into N element can be obtained by calcination at the temperature of 450 °C. Lastly, there is also a weak peak at 2368 cm$^{-1}$ appearing in the spectra of all samples that was neither from $TiO_2$ nor from urea. Hence, the peak possibly originated from the laboratory impurity contaminating KBr, which was used for pelleting the samples.

### 2.1.4. By Scanning Electron Microscope (SEM)

In order to investigate the surface morphology of the N-doped $TiO_2$ photocatalyst, their SEM images were taken, which are displayed in Figure 4. The particles of $TiO_2$ are seen clearly as crystalline forms. The less crystalline phase is demonstrated by the images of all N-doped $TiO_2$, and the amorphous form is observed for $TiO_2$–N (15) with highest amount of N dopant. The similar images due to the higher dopant content have also been reported by Khan et al. [19]. The less crystallin phase may infer the incorporation of N into the $TiO_2$ structure [16,21]. This phenomena matched well with the XRD data.

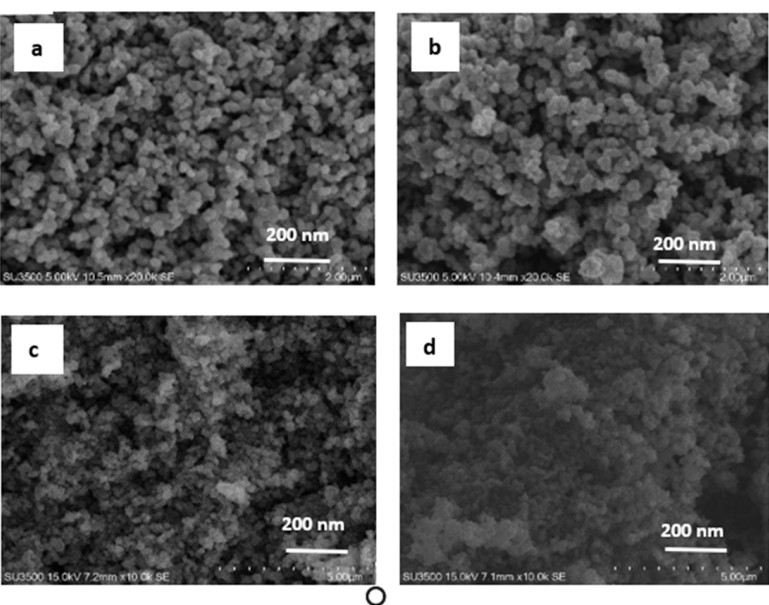

**Figure 4.** The SEM images of (**a**) $TiO_2$, (**b**)$TiO_2$–N (5), (**c**) $TiO_2$–N (10) and (**d**) $TiO_2$–N (15).

*2.2. Photocatalytic Activity $TiO_2$–N under Visible Light in the Removal of Pb(II)*

2.2.1. Influence of N Doping

The activity of the doped photocatalyst was evaluated by applying it to the photocatalysis of Pb(II) under visible light process, as well as under dark and UV irradiated conditions for comparison. The results of the Pb(II) removal are illustrated in Figure 5.

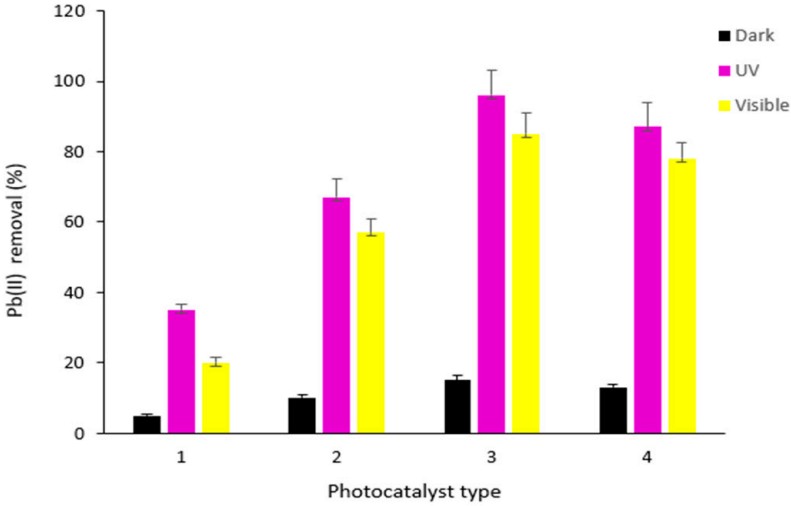

**Figure 5.** The effectiveness of the Pb(II) photo-oxidation over (**1**) $TiO_2$, (**2**) $TiO_2$–N (5), (**3**) $TiO_2$–N (10) and (**4**) $TiO_2$–N (15); under dark condition; and UV and visible light irradiation (photocatalyst weigh = 15 mg, volume of Pb(II) solution = 25 mL, Pb(II) concentration = 15 mg/L, reaction time = 30 min and solution pH = 7).

It can be seen in the figure that N-doped $TiO_2$ posed higher activity in the Pb(II) photocatalytic removal both under visible and UV lights compared to the undoped $TiO_2$ activity. The same trend has also been acquired by many authors [16–20,23,24]. The figure also shows that the Pb(II) ions could be removed under dark conditions, due to the adsorption of $Pb^{2+}$ on the $TiO_2$ surface. It is inferred that the photocatalytic removal is initiated and/or accompanied by the adsorption step [18]. The enhancement of the visible photocatalytic-oxidation was promoted by lowering band gap energy (Eg), that allowed $TiO_2$ to be

activated by visible light generating a lot of OH radicals for oxidation [16–20,23,24]. The reactions of the formation of OH radicals and Pb(II) photo-oxidation were presented as Equation (1), Equation (2) [11–14] and Equation (3) [7–9]. The effective photocatalytic-oxidation of Pb(II) under visible light provides a potential and promising method to be applied on a larger scale for industrial wastewater treatment.

$$TiO_2\text{-N} + h\nu \rightarrow TiO_2\text{-N} + h^+ + e^- \tag{1}$$

$$H_2O + h^+ \rightarrow OH + H^+ \tag{2}$$

$$Pb^{2+} + 2 \cdot OH \rightarrow PbO_2 + 2H^+ \tag{3}$$

$$TiO_2\text{-N} + h^+ + e^- \rightarrow TiO_2\text{-N} + heat \tag{4}$$

Doping N could also considerably improve the photocatalyst activity under UV irradiation, which can be promoted by slower recombination of $e^-$ and $h^+$ (electron and hole) pair, that is, generated during light exposure, as presented in Equation (4). It should be noted that the recombination proceeds naturally and can cause photo-oxidation or reduction reactions. Therefore, inhibition of the recombination needs to be afforded. The recombination can be retarded by doping, since the N dopant can act as electron-hole separation center [17,19,21] by capturing the electrons, which can prevent the recombination. Hence, doping N atom essentially narrows the band gap of $TiO_2$ for the photo-excitation or red shift and simultaneously delays the recombination rate of photogenerated electron–hole pair. The synergic role gives rise to the high effective photo-oxidation.

Moreover, the photocatalytic oxidation of Pb(II) under UV light is seen to be more effective than that of under visible light. The Eg of $TiO_2$ is 3.2 eV that is equal to UV light, enabling $TiO_2$ to be activated by UV irradiation, and hence more OH radicals could be provided. In contrast, the energy of visible light t is lower than the Eg of $TiO_2$, consequently $TiO_2$ was less active in generation of OH radicals [19], which resulted in the lower photo-oxidation.

Figure 5 also illustrates that higher amount of N-doped raised the photo-oxidation efficiency and reached maximum at 10% of N. The improvement was promoted by increasing their activity under visible light due to the lower Eg. Additionally, more N dopant content may improve the retardation of the recombination, further enhancing the Pb(II) oxidation. On the contrary, at the N content beyond its optimum value, the Pb(II) oxidation seems to be detrimental. The excess of N dopant (15% N) considerably decreased the crystallinity of $TiO_2$, which extended the amorphous phase, as described above. The greater amorphous portion in $TiO_2$ structure could adsorb more water molecules that prevent the formation of OH radicals [19]. It is also possible that the residue of urea in $TiO_2$–N (15) covered the active sites on the $TiO_2$ surface, thereby inhibiting the OH radical formation. These conditions explain the decrease in the photo-oxidation. Same finding was also found by some studies [16–20]. Based on their Eg values, 15% of N-doped photocatalysts posed lowest Eg, suggesting the highest visible light absorption, and this showed the highest photo-oxidation effectiveness. In fact, the highest photo-oxidation result was shown by $TiO_2$–N with 10% of N content. It is therefore obvious that both Eg value and the content of N dopant played role in the photocatalysis process. In this case, the N content exhibited higher role than the Eg value did.

## 2.2.2. Influence of Irradiation Time

As seen in Figure 6, prolonged irradiation time up to 30 min could considerably improve the photo-oxidation, but upon further expansion of the irradiation time longer than 30 min, a plateau of the photo-oxidation is observed. The similar trend data was also noted previously [12,14,15,23,25].

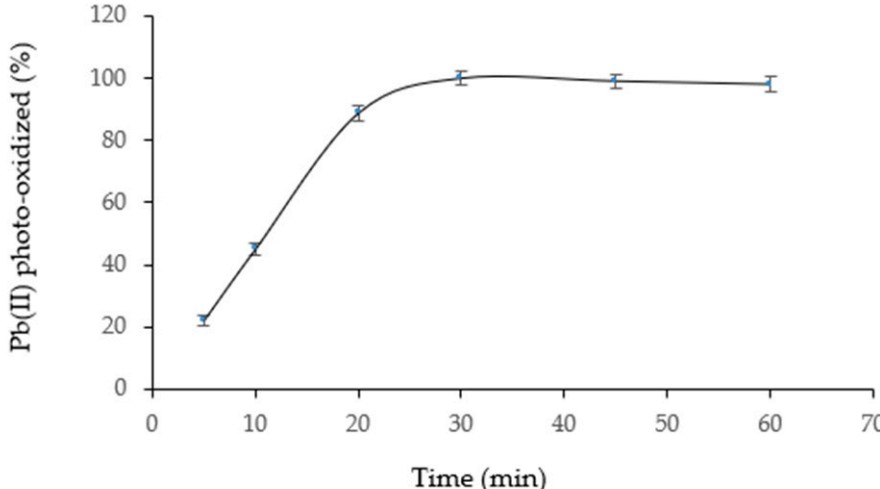

**Figure 6.** The influence of the irradiation time on the Pb(II) photo-oxidation (photocatalyst weight = 15 mg, volume of Pb(II) solution = 25 mL, Pb(II) concentration = 15 mg/L, and solution pH = 7).

With the extended time, the contact between the light and $TiO_2$ proceeded to be more effective, producing more OH radicals. In addition, with the prolong time, a greater collision frequency occurred between the hydroxyl radicals and ion Pb(II) in the solution. This conductive condition, therefore, promoted more effective photooxidation of Pb(II). The longer process than 30 min resulted in a greater amount of the $PbO_2$ deposited on the surface of $TiO_2$–N. Covering the $TiO_2$ surface by $PbO_2$ can limit the visible light tht reaches the active surface of $TiO_2$. As a consequence, the production of the OH radicals could not be enhanced [23], so that the OH radicals available remained same, leading to constant photo-oxidation. Moreover, the short optimum time for the most effective Pb(II) removal is beneficial in terms of application on an industrial scale.

2.2.3. Influence of Photocatalyst Weight on the Photo-Oxidation of Pb(II)

The photo-oxidation of Pb(II) sharply increases with the elevation weight of the photocatalyst, as demonstrated by Figure 7. The increase of the photocatalyst weight provided more OH radicals that could enhance photodegradation.

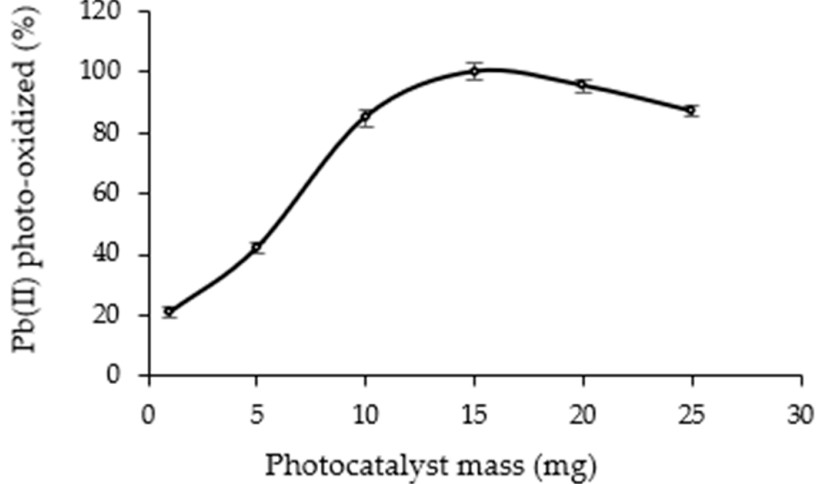

**Figure 7.** The influence of the photocatalyst mass on the Pb(II) photo-oxidation (volume of Pb(II) solution = 25 mL, Pb(II) concentration = 15 mg/L, reaction time = 30 min and solution pH = 7).

This trend data is in perfect accordance with the data resulted by some studies [8,12,14,23]. The larger weight that exceeded the optimum level caused a detrimental in photodegradation. The excessive photocatalyst elevated the turbidity of the mixture that inhibited the light penetration [8,14,23]. The other possible reason is that the large photocatalyst dose allowed for agglomeration that depleted the active photocatalyst surface; thus, less photooxidation efficiency was obtained [12]. The mass producing maximum photo-oxidation is found as 15 mg for 25 mL of the Pb(II) solution, which is equal to 0.6 g/L of the photocatalyst dose. Such a dose is believed to be cost-effective on a larger scale.

### 2.2.4. Influence of Solution pH

The initial pH is one of the most effective parameters in photocatalytic processes, which influences on the adsorption of the substrate on the photocatalyst surface and hence the degradation. The success of the adsorption is dictated by the suitability of the substrate and the photocatalyst surface charges [12]. It is notable in Figure 8, raising solution pH up to 8 was found to noticeably enhance the photo-oxidation. The further elevation of the pH decreased the photo-oxidation.

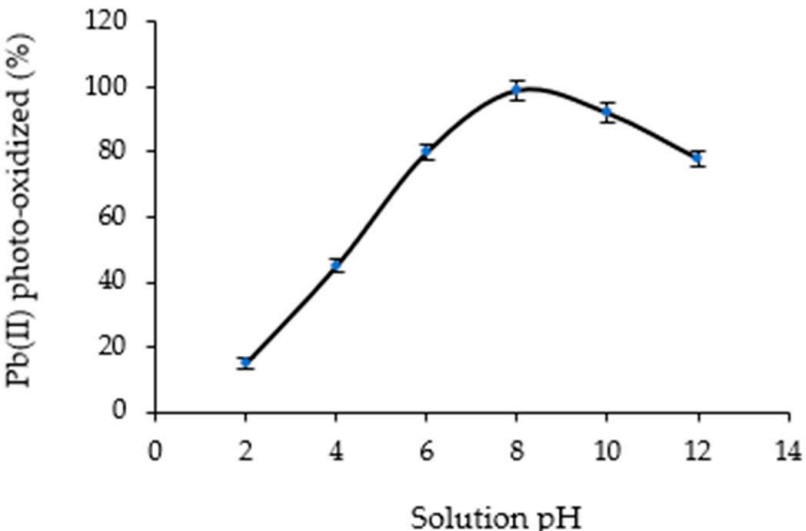

**Figure 8.** The influence of solution pH on the Pb(II) photo-oxidation (photocatalyst weigh = 15 mg, volume of Pb(II) solution = 25 mL, Pb(II) concentration = 15 mg/L and reaction time = 30 min).

In the solution with low pH, the surface of $TiO_2$ was protonated to form positive charge ($TiO_2H^+$), inhibiting the OH radicals formation [8,12,14,23]. On the other side, the cationic of $Pb^{2+}$ was formed predominantly in the solution [7,8], which repulsed the adsorption of $Pb^{2+}$ on the surface of the protonated N-doped $TiO_2$. These simultaneous conditions led to less effective photo-oxidation. When the pH was gradually increased up to 8, the protonation of $TiO_2$ gradually depleted, which generated a greater amount of OH radicals. In such a pH range, a large number of $Pb^{2+}$ were available [7,8], enabling it to interact with $TiO_2$ surface effectively. This condition was very conducive to reaching high photo-oxidation. At pH higher than 8 (base condition), the surface of $TiO_2$ was found as anionic $TiO^-$ species that were inhibited to release OH radicals [8,12,14,23]. In the base solution, $Pb(OH)_2$ precipitate was formed [7,8], which constrained the light penetration, consequently limiting the photocatalyst to produce the OH radicals. The lower number of OH radicals available and the presence of the precipitate explained the significant decline of the photo-oxidation. From the data, it is found that the most effective process took place at neutral pH and has the potential to be applied on an industrial scale.

### 2.2.5. Detection of PbO$_2$ Produced from the Photo-Oxidation

In order to detect the result of Pb(II) oxidation, the SEM-EDX analysis of TiO$_2$–N (10) before and after being used for Pb(II) photo-oxidation was executed, and the SEM images and the EDX spectra were displayed as Figures 9 and 10, respectively. At the SEM image of TiO$_2$–N (10) after being used for photo-oxidation, the small particles (arrow signs) over the surface of TiO$_2$ are observable, which were not seen in the image of TiO$_2$–N (10) before it was used. The particles were very likely of PbO$_2$ resulting from the photo-oxidation of Pb(II).

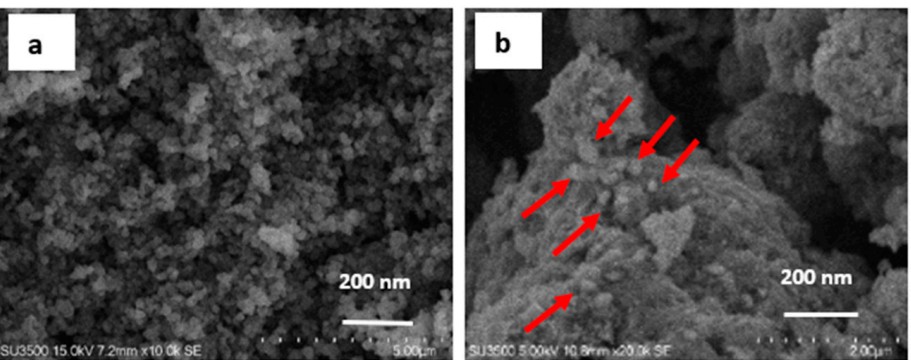

**Figure 9.** The SEM images of (**a**) TiO$_2$–N (10) before and (**b**) TiO$_2$–N (10) after being used for Pb(II) photo-oxidation.

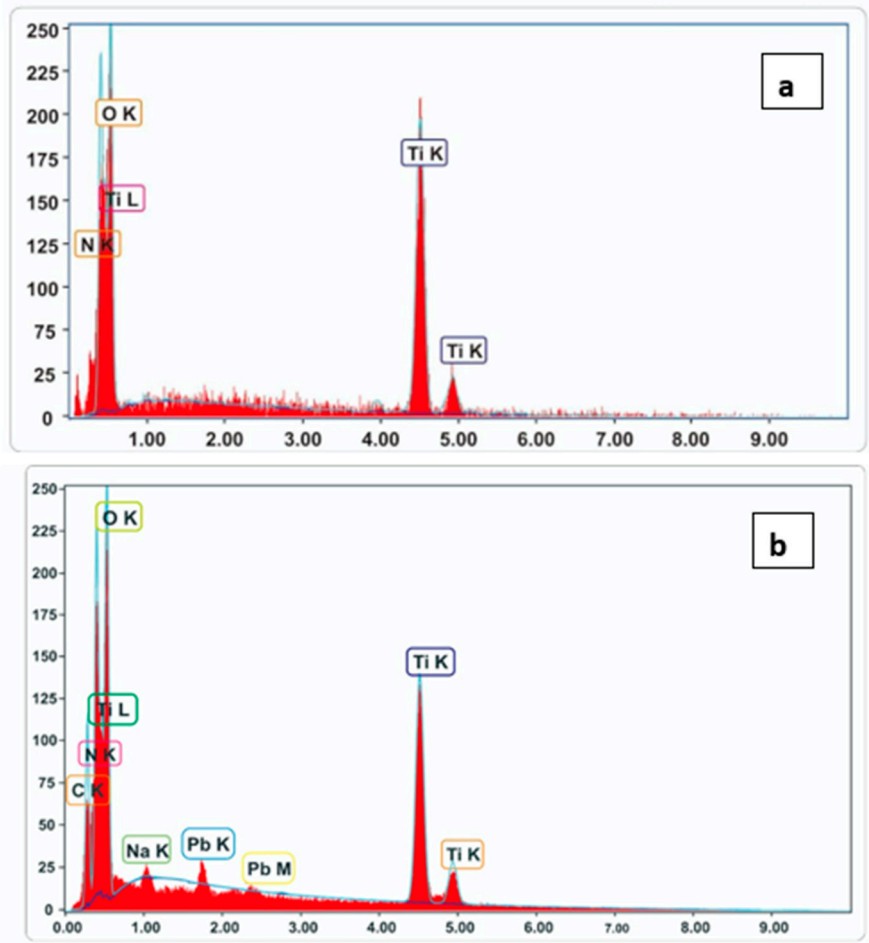

**Figure 10.** The EDX spectra of (**a**) TiO$_2$–N (10) before and (**b**) TiO$_2$–N (10) after used for Pb(II) photo-oxidation.

The presence of $PbO_2$ was strongly supported by EDX spectra of $TiO_2$–N (10) after being used for oxidation, which demonstrates the appearance of Pb peaks. Such peaks were invisible in the spectra of $TiO_2$–N (10) before used for oxidation. Other supporting data is ascribed by increasing intensity of oxygen peak compared to the intensity of oxygen in the spectra of $TiO_2$–N (10) before used. The increase of the intensity assigned the increase of the oxygen amount that could be contributed by $PbO_2$ material. Furthermore, the intensities of Ti, N and C peaks found in $TiO_2$–N (10) after used were notably reduced, due to the incorporation of Pb and oxygen atoms into the photocatalyst.

The quantitative compositions of $TiO_2$–N (10) before and after being used for Pb(II) photo-oxidation that was derived from the EDX spectra by data processor in the machine are displayed in Table 2. Generally, the compositions were consistent with EDX spectra. The appearance of carbon element in both photocatalysts could be due to laboratory impurity and the residue of the incomplete decomposed urea, which are also observed in the respective FTIR spectra. The considerable quantity (5.20%) of Pb formed implied the effective photo-oxidation of $TiO_2$–N (10) under visible irradiation. Hence, there is obvious evidence that the photo-oxidation of Pb(II) yielded of $PbO_2$, by following the reaction in Equation (4). The formation of $PbO_2$ from Pb(II) photo-oxidation well agrees with the finding reported previously [8,9]. The formation of the solid $PbO_2$ is beneficial in terms of solid waste treatment due to the less toxic and handleable waste.

**Table 2.** The composition the doped photocatalyst based on the EDX data.

| Element | The Content of the Element in $TiO_2$–N(10) (% Mole) | |
| :---: | :---: | :---: |
| | **Before Photo-Oxidation** | **After Photo-Oxidation** |
| Ti | 27.10 | 21.40 |
| O | 47.40 | 52.30 |
| N | 15.30 | 11.40 |
| C | 10.20 | 7.60 |
| Pb | - | 5.20 |
| Na | - | 2.10 |

In addition to Pb, the new peak associated with Na element is also observable. The presence of Na could be originated from NaOH solution that was added to elevate pH into 8, from pH 4 of the solution during photo-oxidation. In the solution, NaOH was ionized into $Na^+$ and $OH^-$ and the positive ion of $Na^+$ adsorbed on the surface of $TiO_2$ photocatalyst having more electrons or negative charged surface. The Na was kept to attach on the surface of $TiO_2$–N (10) that was detectable by EDX analysis.

### 2.2.6. The Activity of the Doped-Photocatalyst with the Repetition Used

In order to know the activity of the N-doped $TiO_2$ photocatalyst after repetition use, in the present study the photo-oxidation of Pb(II) under optimal condition over $TiO_2$–N (10), after being used in several times, was observed. The results are presented as Figure 11 and Table 3, exhibiting a gradual decrease of photoactivity with more repetitive use.

The first use of $TiO_2$–N (10) photocatalyst resulted in 24.5 mg/g of the Pb(II) photo-oxidation, which was about 98% toward the initial Pb(II) concentration, referring to the high photoactivity of $TiO_2$–N (10). In the second use of the photocatalyst, an insignificant decrease of the Pb(II) photo-oxidation was obtained—22.63 mg/g, or about 91%. Such photo-oxidation result implied that the activity of $TiO_2$–N (10) photocatalyst remained high after being used once. Additionally, the very low decrease of the photo-oxidation could be effected by of 24.5 mg/g of the Pb occupation on the photocatalyst surface, which still could provide appreciable active surface of the photocatalyst. The appreciable surface active allowed the photocatalyst to have effective contact with visible light, to further produce suffice of OH radicals for the photo-oxidation.

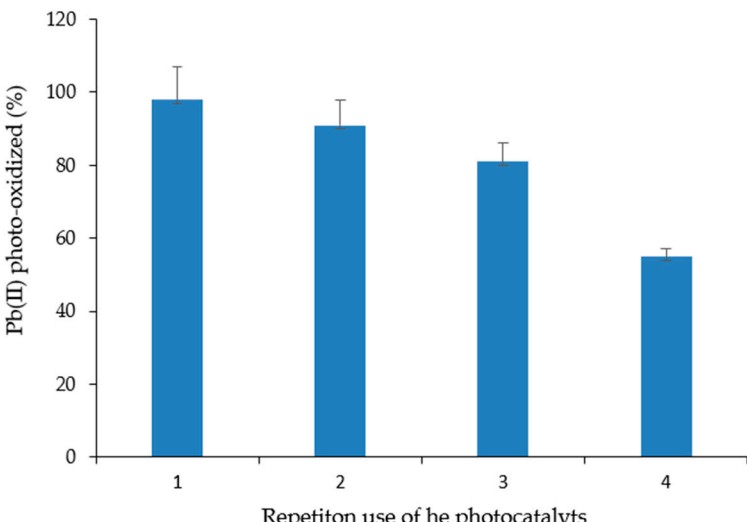

**Figure 11.** The activity of TiO$_2$–N (10) on the Pb(II) photo-oxidation with several repetitions.

**Table 3.** The activity of repeatedly used TiO$_2$–N (10) photocatalyst.

| Repetition of TiO$_2$–N(10) Use | Pb Resulted from the Photo-Oxidation over TiO$_2$–N(10) (mg/g) | Pb(II) Photo-Oxidized (%) | Total Pb Resulted from the Photo-Oxidation Distributed over TiO$_2$–N(10) Surface (mg/g) |
|---|---|---|---|
| 1st | 24.53 | 98.12 | 24.53 |
| 2nd | 22.63 | 90.50 | 47.16 |
| 3th | 20.33 | 81.30 | 67.49 |
| 4th | 13.76 | 55.02 | 81.25 |

The third repetitive use of the photocatalyst in the Pb(II) photo-oxidation yielded 20.30 mg/g or about 81 %, which was less than that that resulted from the second-use photocatalyst. The result of the photo-oxidation (81%) suggested the decent activity of the third-use photocatalyst. The slight depletion of the photo-oxidation could have been generated by the photocatalyst that was covered by 47.13 mg/g of the total Pb. This covering surface of the photocatalyst was believed to keep adequate surface area of the photocatalyst, which could effectively keep contact with the visible light to produce adequate number of OH for photo-oxidation.

When the photo- oxidation of Pb(II) process was conducted by applying the third-use photocatalyst through fourth process, the photo-oxidation drastically declined into 13.76 mg/g or around 55%. This low photo-oxidation result showed that the photocatalyst suffered from the activity. This significant decrease of the photo-activity was induced by smaller surface area of TiO$_2$–N (10) because of the covering by large Pb about 67.43 mg/g. Such noticeable covering obviously prevented the interaction between the photocatalyst with the visible light, which resulted in lack of OH radicals. From the data, it is clear that the repeatedly used photocatalyst up to three times showed significant activity in the Pb(II) photo-oxidation. Hence, the photocatalyst is believed to have promise in terms of application on an industrial scale.

## 3. Materials and Methods

### 3.1. Materials

The chemicals used in this research were $TiO_2$, urea, $Pb(NO_3)_2$, HCl and NaOH, which were purchased from E. Merck with analytical grade and were used without any purification.

### 3.2. Methods

#### 3.2.1. Doping Process of N on $TiO_2$

Doping was performed by hydrothermal method reported previously [23] with small modification. $TiO_2$ powder (about 1 g) was dispersed in 200 mL of 1 g/L urea solution in water solvent. The mixture was placed in autoclave and then was heated at 150 °C for 24 h. The doped $TiO_2$ -N that resulted was dried at 100 °C for 30 min and continued by calcination at 400 °C for 2 h. The sample was kept for characterization and activity evaluation. With such an amount of urea, theoretically, N content in $TiO_2$ was about 5 % w. The same procedure was repeated for urea with 2 g/L and 3 g /L of the concentrations, giving approximately 10% w and 15% w of N content in the doped $TiO_2$ respectively. Therefore, the doped photocatalyst samples were coded as $TiO_2$–N (5), $TiO_2$–N (10) and $TiO_2$–N (15).

#### 3.2.2. Characterization of N-Doped $TiO_2$

The doped photocatalysts obtained were characterized by using Pharmaspec UV-1700 Diffuse reflection ultra violet (DRUV) spectrophotometer to determine their band gap energies (Eg). The DRUV spectra were recorded in the wavelength ($\lambda$) range of 200–700 nm. The values of Eg were calculated based on the wavelengths of absorption edge by following relationship of Eg = $1240/\lambda$ [20]. The wavelengths of absorption edge were estimated based on the intersection of the straight lines of Y-axis with X-axis [20].

A Shimadzu 6000X-XRD machine with radiation source of Cu K$\alpha$ (1,54056 Å) was operated at 30 mA of the current and 40 kV of the voltage to detect the crystallinity of the samples. The XRD patterns of the samples, having about 200 μm of the particle size, were scanned in the range of the 2 tetha of 5–80° with scanning rate was 5°/min. Fourier transform infrared (FT-IR) spectra with the wavenumber of 4000–400 cm$^{-1}$ of the KBr pelleted samples were recorded on Prestige 21 Fourier-Transform Infrared spectrophotometer. From the FTIR spectra, the characteristic chemical bond vibrations could be found. In order to get the surface morphology of the samples, the SEM-EDX images of the samples were taken by Hitachi SU 3500 Scanning Electron Microscopy and Energy Dispersive X-ray (SEM-EDX) equipped with Coating Hitachi MC1000 ION SPUTTER 15 mA and 20 s. In this analysis, the samples were initially metalized by gold coating. All the instruments used are available at Gadjah Mada University, Yogyakarta, Indonesia.

#### 3.2.3. Photo-Oxidation of Pb(II) in the Solution over $TiO_2$–N Photocatalyst

The photo-oxidation of Pb(II) was conducted by batch technique in the apparatus seen in Figure 12. The Pb(II) solution 15 mg//L 25 mL in a beaker glass was mixed with 15 mg of $TiO_2$–N (5), and the glass was put in the photocatalysis apparatus. Next, the beaker glass in the apparatus was irradiated with three wolfram lamps Philip 20 watt as source of visible light, accompanied by magnetically stirring with 200 rpm of the stirring rate for 30 min. The Pb(II) left in the solution was analyzed by Perkin-Elmer 110 AAS machine. The concentration of the Pb(II) was determined by extrapolation on the corresponding standard curve. The same procedure was copied for processes under dark condition, under UV irradiation emitted from 20 watt black light blue (BLB)-type UV lamps, for process with $TiO_2$–N containing N of 10% and 15%, and for processes with various irradiation times (5, 10, 20, 30, 45 and 60 min), photocatalyst weights (1, 5, 10, 15, 20 and 25 mg) and solution pH values (4, 6, 8, 10 and 12). When one parameter was varied, other parameters were kept to be constant. In addition, the repetitive use of the doped photocatalyst in the photo-oxidation was also preceded by following the same procedure. Each experiment

was repeated three times, and the deviation of the photo-oxidation of Pb(II) results were found to be 5–10%.

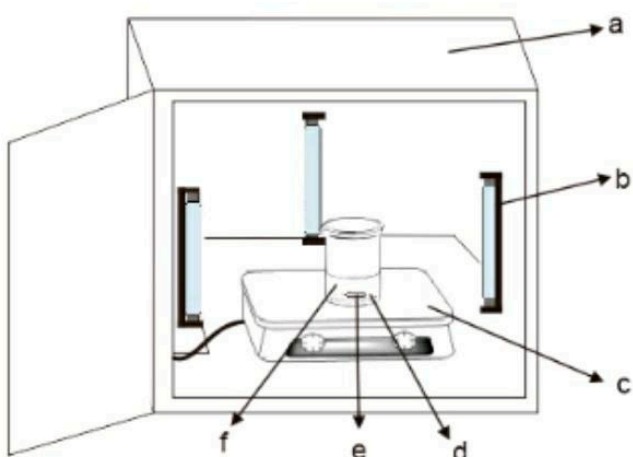

**Figure 12.** A set of apparatus used for Pb(II) photo-oxidation processes composed of: (**a**) melamine box, (**b**) visible or UV lamps, (**c**) magnetic stirrer plate, (**d**) photocatalyst powder, (**e**) magnetic stirrer bar and (**f**) sample solution.

## 4. Conclusions

The N-doped $TiO_2$ photocatalyst was successfully prepared using the hydrothermal method, which narrowed the band gap energy, allowing it to absorb visible light. It was found that doping N on $TiO_2$ structure could improve its activity in the Pb(II) photo-oxidation under visible light irradiation. The most effective Pb(II) photo-oxidation (98%) from Pb(II) 15 mg/L in 25 mL solution was reached by using $TiO_2$–N with 10% of N fraction, by applying the condition of 15 mg of the doped photocatalyst, 30 min and pH 8. It is also evident that $PbO_2$ is produced from the Pb(II) photo-oxidation. The doped $TiO_2$ photocatalyst with three repetitions demonstrated sufficient activity in the photo-oxidation. The low photocatalyst dose, short reaction time and neutral pH used to achieve the highest effective process are beneficial factors allowing the method to be applied for the treatment of industrial-Pb(II)-containing wastewater.

**Author Contributions:** Conceptualization, E.T.W.; methodology, A.S.; research experiment and data execution, T.R. and A.R.H.; data analysis, S.S.; writing—original draft preparation, T.R., A.R.H. and S.S.; writing—review and editing, E.T.W.; All authors have read and agreed to the published version of the manuscript.

**Funding:** This research was funded by Faculty of Mathematic and Natural Sciences Gadjah Mada University through a grant of Public Financial with the contract number 77/J01. 1.28/PL.06.02/2020.

**Acknowledgments:** Authors greatly thank to Faculty of Mathematic and Natural Sciences Gadjah Mada University.

**Conflicts of Interest:** The authors declare no conflict of interest. The funders had no role in the design of the study; in the collection, analyses or interpretation of data; in the writing of the manuscript; or in the decision to publish the results.

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
