# Peer review of "Photocatalysis over N-Doped TiO2 Driven by Visible Light for Pb(II) Removal from Aqueous Media"

_catalysts, doi:10.3390/catal11080945_

Round 1

Reviewer 1 Report

The manuscript submitted by author related to synthesis of N-doped TiO2 photocatalysts for Pb(II) removal under visible region is the good effort made by authors. But manuscript lacks in several ways.

Suggestions and concerns are listed below for author attention.

  1. The first very important issue about the English of this script. This script is very poorly written and presented Please write your text in good English (American or British usage is accepted, but not a mixture of these). English language manuscript require editing to eliminate possible grammatical or spelling errors and to conform to correct scientific English.
  2. Second major concern that I have notice that there are several articles available on TiO2 based photocatalysts for Pb removal. Provide statement describing the one or more key hypotheses that the work described in the manuscript was intended to confirm or refute. Inclusion of a hypothesis statement makes it simple to contrast the hypothesis with the most relevant previous literature and point out what the authors feel is distinct about the current hypothesis (novelty).
  3. Under introduction section authors have mention “ Pb(II) ion is non-biodegradable and tends to accumulate in living tissues, causing various diseases and disorders”. What are the diseases and disorder causes by lead need to be added in the sentences?
  4. Authors have mention “Lead (Pb), along with Cd, Hg, Cr(VI) and As, is categorized as the most toxic of the heavy metal”. Provide the appropriate references to support of this sentences. [Science of The Total Environment, 147851, 2021; Materials Today Chemistry 18, 100376, 2020]
  5. On page 2 , authors have mention “Non-metal dopant has been reported to give more effective in narrowing gap or decreasing Eg of TiO2, compared to the metal dopant, because the size of non-metal  elements are smaller than the metal dopants”. Define Eg and provide the full form of it.
  6. In section “2.1.1. By DRUV method”. Define DRUV?
  7. On page 3, Figure 1. The DRUV Spectra of a) TiO2, b) TiO2-N(5) c) TiO2-N(10), d) TiO2-N(15). Indicate the graph showing Eg (eV) of a) TiO2, b) TiO2-N(5) c) TiO2-N(10), d) TiO2-N(15).
  8. On page 2, author have directly started section “2. Results and Discussion 1. Characterization of TiO2-N photocatalyst” But no information about how the methodology used for synthesis of TiO2-N photocatalyst. Provide the synthesis method of TiO2-N photocatalyst.
  9. What are the chemicals used in the work, there grade, purity, and made need to be added.
  10. Similarly characterization model name, country and operating conditions of each of the instruments used in the script need to be added.
  11. On page 3, section “2.1.2. By XRD method , authors have mention “The XRD patterns of TiO2 and TiO2-N photocatalyts are displayed as figure 2. Sev-98 eral 2θ values of 25.25°, 37.52°, 48.02°, 53.58°, 54.88°, 62.61°, 68.78°, 70.33°, 75.07° and 99 82.68° are observed, which are assigned to anatase TiO2” Provide the JCPDS number to confirm the phase and other details of TiO2 and TiO2-N.
  12. On page 4, Figure 2. The XRD patterns of a) TiO2 (uncalcined), b) TiO2 (calcined at 400 oC), c) TiO2-N(5), d) 104 TiO2-N(10), and e) TiO2-N(15). Denote the peak position along with there respective hkl value.
  13. Authors have mention nitrogen was doped in TiO2, what are the source of N?I cannot find anywhere in this script.
  14. On page 5, Section 2.1.4. By SEM, Figure 4. The SEM a) TiO2, b)TiO2-N (10), and c) TiO2-N(10) after used for Pb(II) oxidation. Provide the clear scale bar. I cannot find any change in the morphology of TiO2-N (10) before and after adsorption of Pb. Author need to again perform the SEM analysis.
  15. Figure 6. The influence of the irradiation time on the Pb(II) photo-oxidation. Provide with error bar in graph.
  16. Similarly Figure 7. The influence of the photocatalyst weight on the Pb(II) photo-oxidation. Provide with error bar.
  17. On page 9, Figure 9. The EDX data of a) TiO2-N before, and b) TiO2-N after used for Pb(II) photo-oxidation. I cannot see any clear EDX. Provide the high quality of EDX plot so that I can co relate the results.
  18. Presentation is very weak. More results and discussion need to be added.
  19. More latest references need to be added in the script.
  20. On page 10, Following section need to be added in the starting of manuscript. After introduction.

“ 2. Materials and Methods

4.1. Materials

The chemicals used in this research were TiO2, urea, Pb(NO3)2, HCl, and NaOH, that  were purchased from E. Merck and were used without any purification.  

4.2. Methods

4.2.1. Doping process of N on TiO2

Doping was performed by hydrothermal method following reported previously [16]. TiO2 powder about 1 gram was dispersed in 100 mL of urea solution 1 g/L. The mixture was placed in autoclave, then was heated at 150 oC for 24 h. The doped TiO2 -N re- sulted was dried at 100 oC for 30 min, and continued by calcination at 400 oC for  2 h. The sample was kept for characterization and activity evaluation. With such amount of urea, theoretically, N content in TiO2 was about 5 %w. The same procedure was re-peated for urea with 2 g/L and 3 g /L of the concentrations, giving around 10%w and  15%w of N content in the doped TiO2 respectively. Therefore, the doped photocatalyst  samples were coded as TiO2-N(5), TiO2-N(10) and TiO2-N(15).

4.2.2. Characterization of N-doped TiO2

The doped photocatalysts obtained were characterized by using Pharmaspec UV-  1700 DRUV spectrophotometer, Shimadzu 6000X-XRD, 8210 FTIR spectrophotometer, and JSM-6510 LA SEM-EDX machines.’

Author Response

Dear Reviewer

Thank you for your comments and suggestions. Please see the attachment

Reviewer 2 Report

Photocatalysis over N-doped TiO2 driven by visible light as a new method for Pb (II) removal

Endang Tri Wahyuni     et al.

The actual presentation, after various readings, did not reach, in the opinion of the reviewer, the necessary standards for publication in “Catalysts”.

Although the subject can be seen as interesting, the way the work was performed and the deepness of results analysis are far from complete.

On a qualitative presentation base, the used language is not easy to understand as it is far from established English; the caution to present data and results is limited, the presentation of the literature (at least 8 different manners to write a reference, with some elements missing) and its use to sustain or support discussion is too simplified when not simply wrong (see for example line 268, the cited reference [16] is not associated to the preparation of N doped titania).

Some more comments and suggestions are following, but the actual project is not acceptable under the present form.

Some scientific comments

Experimental part

Line 268 – the reference [16] deals with TiO2 “doped” with Ag and not N.

Line 269 – what is the solvent of urea?

Line 276 – Apparently, 3 “doped” catalysts were prepared, the higher N content being 15. But EDX analysis was performed with a sample containing more “N” concentration.

Line 278 – the “characterization” section is limited to the name of used set-ups: nothing was written on the way the samples to be characterized were prepared (XRD, FTIR, SEM-EDX)

Line 285 - The text indicates that the reaction occurred in an Erlenmeyer, whereas the apparatus scheme of Figure 10 is presenting a beaker.

Line 287 – the text speaks of a lamp, and the figure 10 shows at least 3 drawings looking like lamps.

In that section, no idea is given on the rate of reacting medium agitation.

Results and discussion

Table 1: How were measured the λ values in Table 1, and what is the accuracy of these values. Consequently, what is the accuracy of band gap?

Figure 2: Figure 2 must contain indexes of diffraction lines. In fact, for the sample with the higher N index (15) a deviation of base line is apparent, between 22 and 30 ° 2θ, generally observed with amorphous materials such as high surface area of silica or active carbon. Then a large fraction of crystalline TiO2 is probably destroyed, but the destroyed phase is not characterized. At least, BET area and porosity are needed to try to understand what occurred at high N “doping” concentrations.

Line 110: Mahy et al. used nanoparticles of TiO2, with properties probably different from those used in the present work. And they did not report strong amorphization of their sample doped with urea.

Lines 114-115: “In the figure 3, it is seen that the FTIR spectra of all TiO2-N samples are similar to that of un-doped TiO2.  

Such statement of the authors is erroneous. The samples with high N content show a NEW Band at 2160 cm-1, that is not mentioned nor discussed by the authors and possibly linked to some C-N chemical links.

SEM: the quality of the graphs observed by referee is poor, and the data contained in the micrographs cannot be red. Therefore, it is difficult to have an idea of micrograph magnification. Further, to be sure that some PbO2 is visible on Figure 4c, it is necessary to perform selective imaging with the SEM, technique not used here, although the equipment had this possibility.

Line 142 and following

Whereas not commented in the experimental part, it seems that two types of lights were used, UV one and visible one. But the reader has no information on the origin and the quality of these light sources. Figure 5 presents various results, but the reader had no idea on the exact experimental conditions used (T, pH, time of reaction, spontaneous Pb (II)/Pb (IV) oxidation – seen through blank experiment, fraction of Pb extracted from solution by adsorption, balance between Pb (II) and Pb (IV), catalyst mass and so on. Therefore, before entering theoretical discussion, detailed comments on experimental conditions seem necessary. The fact that “N” content of 10 seems an optimum deserves specific comments, associated with reaction medium turbidity measurements and better characterization of the three doped samples before and after reaction.

Line 206 – Figure 6. As before, error bar is not given to verify if a true plateau was obtained or is unknown chemistry transformed back Pb (IV) to Pb (II). For the reviewer, a plateau is obtained and obviously, nothing else can appear.

Line 223 and following, when increasing pH, (probably through NaOH addition), a carbonatation of the suspension can also exist, changing the “reactivity” of Pb species versus experimental conditions. Fine chemical analyses seem necessary.

Line 241 and following: Detection of PbO2.

The quality of Figure 9 did not allow to identify all lines shown; further, the quality of base line between the solid samples before and after Pb “elimination” is quite different, a fact not commented.

The results of Table 2 must be better analysed. First, a rather important amount of C appeared: what is its origin. Does the possibility of both N and C doping exist in the TiO2 samples or does C come from a “laboratory contamination”? The atomic O/Ti ratio must be close to 2. Was this value verified?

The samples used for SEM EDX contain more “N” than those used for other characterizations. By reference to Figure 5, such a sample must be less active than the sample TiO2-N10, and therefore the possible presence of PbO2 could be due to other reasons than a good activity. The authors must clarify this point. The conditions of use of the samples presented in EDX results must also be clarified (reaction T, time, pH, catalyst weight).

To summarize,

Among other points, the authors

1/ must give an idea of the repeatability of their catalytic measurements, and define simple sorption of Pb on the catalyst surface-porosity, without illumination

2/ use the same catalysts for all characterizations showing also Surface area and porosity determination

3/ analyse better the IR and UV results

4/ detail better the experimental procedures

5/ make normalized literature presentation

6/ Find a scientific English native to optimize the writing form

Author Response

(The authors gave the same response as above.)

Round 2

Reviewer 1 Report

Most of the comments were revised by authors. But still few concern are there with this script, which need to be revised.

Find the comments below

  1. On page 6, section 2.1.4. By Scanning electron microscope (SEM), Figure 4. The SEM images of a) TiO2, b)TiO2-N (5), c) TiO2-N(10), and d) TiO2-N(15). The scale bar is not visible. Provide the clear scale bar.
  2. On page 7, Figure 5. The effectiveness of the Pb(II) photo-oxidation over : 1) TiO2, 2) TiO2-N(5), 3) TiO2-N 221 (10), and 4) TiO2-N (15), under dark condition ,and UV and visible light irradiation (Photo-222 catalyst weigh = 15 mg, volume of Pb(II) solution = 25 ml, Pb(II) concentration = 15 mg/L, reaction 223 time = 30 min and solution pH = 7). Kindly provide with error bar.
  3. On page 10, section , 2.2.4. Influence of solution pH, no justification and complete discussion about this section. More elaborate discussion are needed with appropriate references.
  4. Page 11, Figure 9. The SEM images of a) TiO2-N(10) before, and b) TiO2-N(10) after used for Pb(II) photo-346 oxidation. Provide with clear scale bar.

Author Response

Thank you for your valuable comments and suggestions. I have tried to revised maximally as suggested 

Reviewer 2 Report

Reviewer thanks the authors for their efforts in attempting to make their contribution more meaningful. But, anyway, the presentation is still difficult and even impossible to accept in its second version.

Reviewer will present some specific points to be considered, these points being only a few among all those that must be corrected.

Results and discussion

Lines 105-106, 111-113:  better in the experimental part

Lines 115-116 difficult to understand,

Lines 135-136: ten 2θ values but only 4 lines with indexes in the XRD figure; be homogeneous.

Lines 181-184: the band at 2160 cm-1 is too far from that at 2052 cm-1, to be attributed to the same type of species. These bands are not present in the urea spectrum, contrary to what was said. IN a general way, bands between 2100-2300 cm-1 are often associated with C≡C, C≡N, N=C=O, N=C=S, N=C=N; such types of chemical links must be tested and or verified

Figure 4: The micrographs are not associated with their magnification and the quality of micrographs does not allow the reader to see what is written on the micrograph legend. Therefore, it is difficult to accept the comments of the authors. Furthermore, it is common that before observing solid samples through SEM, these samples are “metallized” with C or gold to allow a good electron imaging. Nothing is detailed in the text except that “Coating Hitachi MC1000 ION SPUTTER 15mA and 20s” seemed to have been used.

Line 209: a fraction of Pb (II) can be simply adsorbed. But the adsorption is quantitatively linked to the surface area of TiO2. However, the specific surface areas of the various TiO2 samples are not known, neither before nor after N introduction. Although the authors have no access presently to such measurements, some literature data can probably give an idea of what happened in term of texture when increasing N content.

Line 226: The writing of TiO2-N is strange: it suggests that N is inserted in TiO2, without exchange between O and N, situation not verified. When metals like Ni, Pt, Pd and others are supported on TiO2, a thermal treatment at around 400°C can completely suppress the accessibility of the metals to reactant in what was called “SMSI” strong metal support interaction. In that case some oxygen of TiO2 was lost. A similar situation may exist in case of N and TiO2 interaction.

Photocatalytic results

Reviewer is still questioning the error bars: how were they estimated? Through theoretical calculations or through experimental repetition. Nothing is indicated in the text. For referee, experimental results from such type of experiments give generally a larger dispersion of results than what was presented.

Lines 278-283 and Figure 6: If after 30 min reaction all Pb (II) was transformed to PbO2, what chemistry can explain that for longer reaction times, some PbO2 can be dissolved again in the reaction medium, as the error bars suggest that the plateau is no more observed.

Figure 9: the same problems as in Figure 4: impossible to accept the statement of the authors without knowing the true magnification of micrographs a and b, that seemed different. What can be seen at higher magnification can escape at minor magnification. The necessity to use localized EDX analysis or to perform a true mapping is obvious.

Figure 10: the values presented on x and y axes cannot be seen. Between the lines PbK and OK, a line appears with a rather important intensity. This line is not attributed to an element: why? Was this line included in the data analysis Table 2?

Table 2: whereas in the photocatalytic experiments all values are given in mg/L or mg/g, in Table 2 results are given in moles. Through a rapid estimation, it is possible to verify that in the experimental conditions of Figure 6, in term of moles, we have some 3 moles of Ti for 1 mole of Pb. If all Pb (II) is eliminated from the reaction medium and is trapped onto the catalyst, the molar ratio Ti/Pb must be 3 and not 4 as given in Table 2. How such a situation can be explained.

How can be explained the strong presence of C and the fact that urea was not fully destroyed after 2 h at 400°C: some literature data on thermal and catalytic decomposition of urea would be necessary to better explain the C presence.

Finally, in photocatalytic reaction in aqueous medium, it is important to check the reuse of catalyst after reaction. No try was described in the actual study, an absence that would not sustain a possible use of this “technology “ on an industrial level.

Author Response

Thank you very much for your valuable comments and suggestions. We have tried to revise and add additional data as suggested asked. We have done our best. we hope our revised article can be accepted to be further published 

Round 3

Reviewer 1 Report

The authors have satisfactorily revised the script.

Author Response

I greatly appreciate for your valuable comments and suggestions to allow our article much better. I do ho hope after 3 times reviewed and revised, our article can be accepted to be further published

Reviewer 2 Report

The third version of the publication draft showed an increase in scientific content, although a deeper analysis of presented results should be necessary. Apparently, only one over five coauthors tried to answer rewiever comments and tried to optimize the discussion.

Among the points,  one of the EDX peaks at around 1 keV is still not commented, whereas it could be due to Na (but of what origin?). Finally, if small PbO2 particles are formed as proposed from SEM micrographs images (without local analyses), their contribution to a decrease in the number of "active sites" poisoning must be very limited compared to the whole disponible surface. Therefore the rather important decrease in efficiency for the second, third and forth reuses is not fully justified.

The quality of english langage is below minimum standard.

Author Response

Thank you very much for your significant comments and suggestions to allow our paper to be much better.  We do hope our article that has been 3x revised, will be accepted to be further published
